# Exhausted CD8+ T cells exhibit low and strongly inhibited TCR signaling during chronic LCMV infection

Ioana Sandu [1,2], Dario Cerletti [1,2], Manfred Claassen[2,3] & Annette Oxenius [1✉]

Chronic viral infections are often associated with impaired CD8+ T cell function, referred to as exhaustion. Although the molecular and cellular circuits involved in CD8+ T cell exhaustion are well defined, with sustained presence of antigen being one important parameter, how much T cell receptor (TCR) signaling is actually ongoing in vivo during established chronic infection is unclear. Here, we characterize the in vivo TCR signaling of virus-specific exhausted CD8+ T cells in a mouse model, leveraging TCR signaling reporter mice in combination with transcriptomics. In vivo signaling in exhausted cells is low, in contrast to their in vitro signaling potential, and despite antigen being abundantly present. Both checkpoint blockade and adoptive transfer of naïve target cells increase TCR signaling, demonstrating that engagement of co-inhibitory receptors curtails CD8+ T cell signaling and function in vivo.

[1] Institute of Microbiology, ETH Zurich, Vladimir-Prelog-Weg 4, 8093 Zürich, Switzerland. [2] Institute of Molecular Systems Biology, ETH Zurich, Otto-Stern-Weg 3, 8093 Zürich, Switzerland. [3]Present address: Internal Medicine I, University Hospital Tübingen, Faculty of Medicine, University of Tübingen, Otfried-Müller-Straße 10, 72076 Tübingen, Germany. ✉email: oxenius@micro.biol.ethz.ch

LCMV (lymphocytic choriomeningitis virus) infection in mice is a well-established model for chronic infection, leading to T-cell exhaustion following chronic infection in mice[1]. Compared to effector CD8 T cells generated following an acute infection, exhausted CD8 T cells have impaired effector functions and are transcriptionally different[2]. An important feature of exhausted CD8 T cells is the co-expression of multiple co-inhibitory receptors (such as PD-1, CTLA-4, LAG-3, TIGIT, CD39, TIM-3), which dampen T-cell activation[3–10] by various mechanisms. These include limiting co-stimulation by receptor competition (CTLA-4, TIGIT[10]), direct inhibition of signal transduction downstream of T-cell receptor (TCR) engagement by limiting the phosphorylation of signaling molecules, such as CD3, ZAP70, and PCK (PD-1[11], TIM-3[12]), restraining metabolic changes[13,14], changes at the transcriptional level (PD-1[15]), interfering with proliferation (LAG-3[16]), or suppressing inflammatory cues (CD39[5]). Significant effort has been invested to enhance the effector functions of exhausted cells; indeed, checkpoint inhibitors, targeting various of the above-mentioned co-inhibitory receptors, are very efficient in improving CD8 T-cell numbers and effector function of exhausted CD8 T cells in both cancer and chronic infections[9]. The phenotypic and functional landscape of exhausted cells is very diverse[17–20], translating into differential responsiveness to checkpoint inhibition. A specific subset of non-terminally exhausted cells, termed memory-like and characterized by expression of TCF1 and SLAMF6[21], was shown to replenish the pool of terminally exhausted cells and to respond to checkpoint blockade by proliferation and differentiation into more effector-like, and eventually terminally exhausted cells[17,21,22]. Despite the compelling evidence that CD8 T-cell function is impaired in chronic LCMV infection and that continued exposure to antigen significantly contributes to exhaustion[23], there is little insight into how much TCR signaling is actually ongoing in exhausted CD8 T cells in vivo during established chronic infection.

Here, we characterize in vivo TCR signaling in virus-specific CD8 T cells in the setting of chronic LCMV infection. We use virus-specific TCR transgenic CD8 T cells expressing the Nr4a1-GFP reporter as proxy for TCR signaling in adoptive transfer experiments. Despite abundant availability of antigen in the form of peptide-MHC class I complexes, there is very limited TCR signaling ongoing in exhausted CD8 T cells during chronic infection, evidenced by low expression of the GFP reporter and by RNAseq analysis of TCR signaling-associated genes. We observe enhanced TCR signaling after in vivo blocking of PD-1/PD-L1 interaction or in vivo exposure of exhausted CD8 T cells to antigen on naive target cells, which have not been exposed to the inflammatory milieu of chronic infection. This observation suggests that the engagement of co-inhibitory receptors, such as PD-1, exerts a pronounced inhibition of TCR signaling in vivo.

## Results

**RNAseq indicates low TCR signaling in chronic infection**. Previous studies showed that, compared to classical effector CD8 T cells formed in acute infection, exhausted virus-specific CD8 T cells are transcriptionally distinct and fail to exert several effector functions, such as production of the inflammatory cytokines IFN-γ and TNF-α[2]. To test whether this defect is due to intrinsically impaired TCR signaling, we set out to investigate whether ex vivo restimulation of these cells in absence of the chronic environment leads to upregulation of new transcriptional modules. CD45.1+ TCR-transgenic P14 CD8 T cells recognizing the LCMV-derived gp$_{33–41}$ peptide were adoptively transferred into CD45.2+ hosts one day prior infection with a low (200 ffu) or high ($2 \times 10^6$ ffu) dose of LCMV clone 13 to induce an acute or chronic infection. Two weeks

later, P14 cells were isolated from the spleens and restimulated in vitro with plate-bound anti-CD3 and anti-CD28 antibodies.

As described previously[2], cells isolated from chronic infection have a distinct transcriptional profile compared to the ones isolated from an acute infection (Fig. 1a). Upon in vitro restimulation in presence anti-CD3 and anti-CD28 antibodies, there were significant changes in CD8 T cells isolated from both infection conditions, especially regarding inhibitory receptor genes, such as Pdcd1 (encoding PD-1) and Cd160, and cytokine production-related transcripts like Ifng, Tnf, Il10, and Xcl1, demonstrating that exhausted CD8 T cells are not inherently unable to transmit TCR signals into a transcriptional output. In fact, 429 genes showed similar changes in expression in both CD8 T cells isolated from acute infection and from chronic infection. Surprisingly, only few genes (35) were differentially expressed in exhausted cells when compared to effector/memory cells after stimulation (Fig. 1b), most of which include inhibitory receptor genes that were already found at higher levels before stimulation and were, therefore, not increased as strongly as in effector/memory cells. Moreover, genes involved in TCR signaling, such as Nr4a1 and TCR-induced genes Ifng and Tnf had a low expression in chronic infection ex vivo, which increased after antibody stimulation, suggesting that the cells were either not properly activated and/or strongly inhibited in vivo.

**NUR77 is downregulated rapidly in P14 cells in vivo**. In order to investigate the TCR signaling in an in vivo setting, we took advantage of a transgenic mouse model, in which GFP is expressed under the control of the Nr4a1 promoter[24]. NUR77, encoded by Nr4a1, is triggered by antigen stimulation in T cells and has been used as a proxy for TCR signaling[24–26]. In vitro stimulation of transgenic CD8 T cells labeled with a cell proliferation dye (CPD) with plate-bound anti-CD3 and anti-CD28 antibodies for 20 h led to fast upregulation of GFP. After the stimulus was removed, the signal was rapidly lost: after approximately 50 h, the GFP levels were similar to the ones observed in naive cells (Supplementary Fig. 1a). Moreover, GFP was downregulated before cells started dividing (29 h after stimulation), suggesting that the initial drop in the observed signal was due to transcriptional regulation and protein degradation, but not proliferation-related dilution. We further sought to characterize the TCR signaling in vivo in chronic LCMV infection. Nr4a1-GFP mice were crossed to the P14 strain to generate P14-Nr4a1-GFP mice. CD45.1+ P14-Nr4a1-GFP CD8 T cells were adoptively transferred into wild-type CD45.2+ C57BL/6J mice 1 day prior infection with a high dose of $2 \times 10^6$ ffu LCMV clone 13. Mice were sacrificed at different time points after infection to investigate TCR signaling ex vivo by measuring GFP in P14 cells. Only 1 day after chronic infection, P14-Nr4a1-GFP cells isolated from the spleen were highly activated (Fig. 2a). Consistent with this observation, the activation marker and inhibitory receptor PD-1 was also upregulated within 24 h after infection (Fig. 2a). However, the GFP levels were rapidly downregulated in vivo already 3 days post infection and kept decreasing with time, despite the maintenance of high PD-1 levels (Fig. 2a–c). Since LCMV has a broad tropism and induces a systemic infection[27], P14-Nr4a1-GFP cells isolated from various infected tissues (lymph nodes, bone marrow, spleen, blood, liver, lung, and kidney) were also analyzed 3 weeks post LCMV infection. Nr4a1-GFP levels were extremely low in all screened tissues 3 weeks into chronic infection compared to in vivo primed cells 1 day after infection (Fig. 2d and Supplementary Fig. 1b–d).

**In vitro stimulation of P14 cells increases TCR signaling**. There is extensive work showing that exhausted CD8 T cells have a

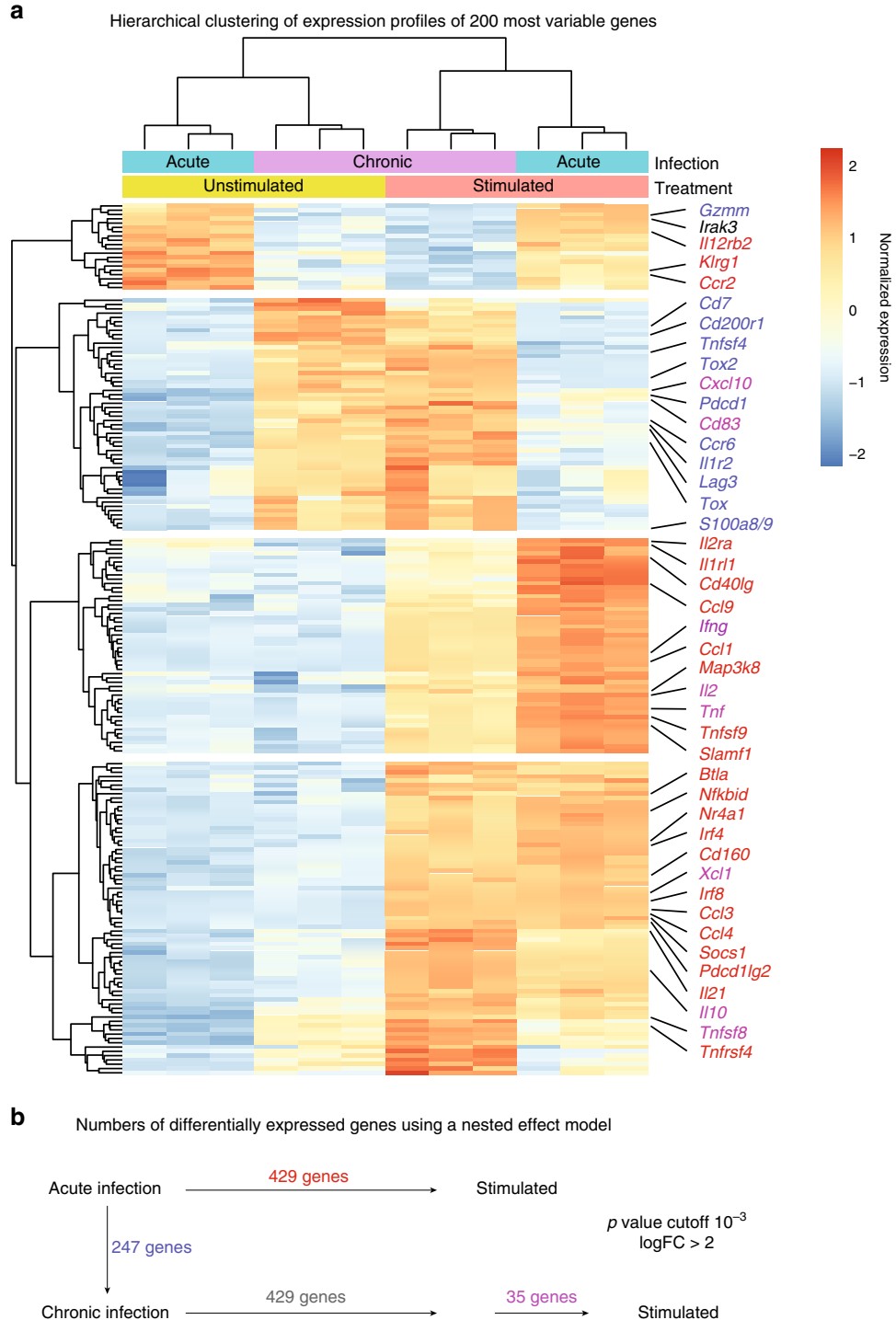

**Fig. 1 Transcriptional profiling of functional or exhausted P14 cells with or without restimulation.** P14 cells were adoptively transferred into mice 1 day prior high or low-dose LCMV clone 13 infection. Animals were sacrificed after 14 days. CD8[+] P14 cells were stimulated with anti-CD3 and anti-CD28 for 4 h. RNA was extracted and sequenced. **a** Heatmap of the 200 most variable gene profiles was generated using hierarchical clustering ($k = 4$). Selected genes are annotated. Heatmap colors indicate normalized gene expression ranging from high (red) to low (blue). Gene color highlights differentially expressed genes between groups indicated in **b**. **b** Number of differentially expressed genes between the conditions are indicated. The nested effect model disentangles the effect of chronic infection and stimulation and contains an interaction term for stimulation in chronic infection. Color code of gene names indicates the group of the differential expression analysis (see Methods for details). Data from one experiment are shown. Source data are provided as a Source Data File.

dysfunctional transcriptional network compared to functional cells developed upon an acute infection[2]. To test whether TCR signaling can be increased in exhausted cells, P14-*Nr4a1*-GFP cells were isolated 3 weeks post chronic or acute LCMV infection, sorted, and stimulated in vitro with anti-CD3 and anti-CD28

antibodies for 6 h. There was a strong upregulation of *Nr4a1*-GFP in cells that responded to stimulation (CD107a[+]) compared to the ones that did not (CD107a[−]) for both P14 cells isolated from acute or chronic infection (Fig. 3). However, the extent to which the *Nr4a1*-GFP signal could be induced in cells isolated from

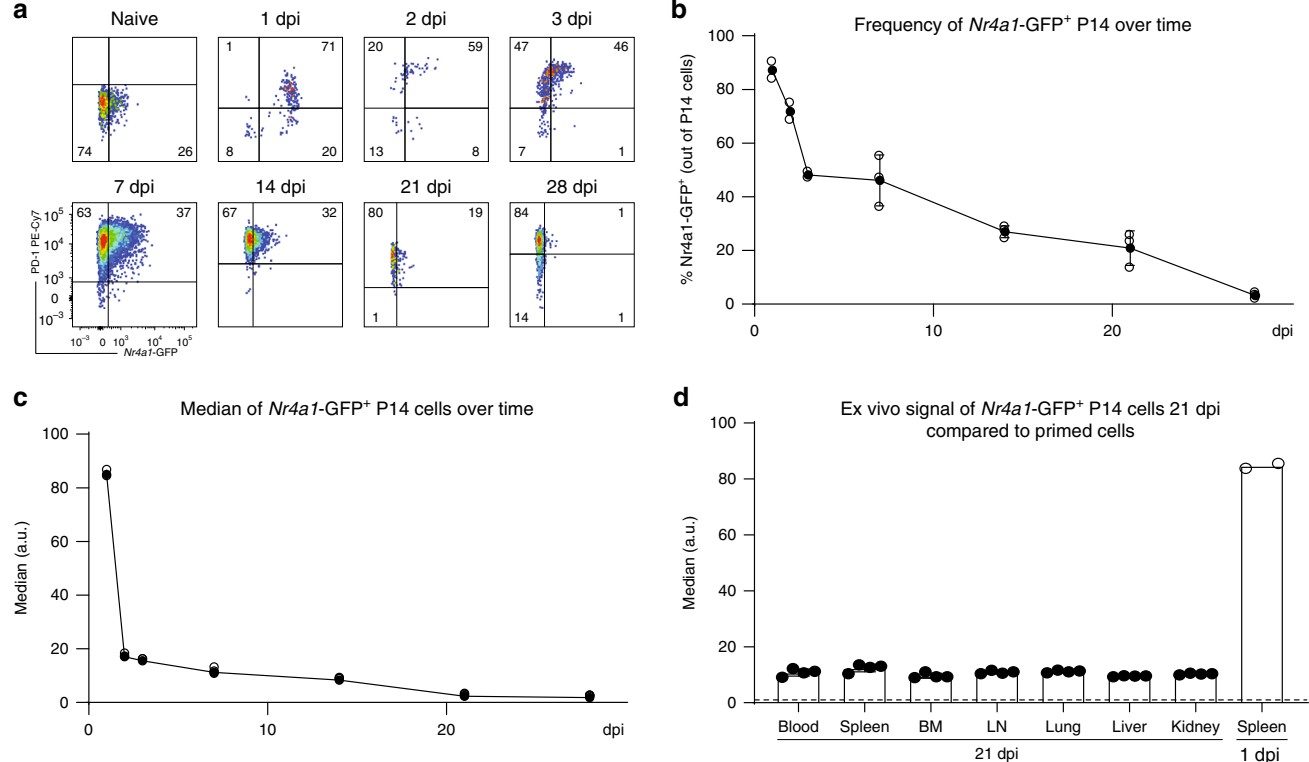

**Fig. 2 In vivo kinetics of TCR activation during chronic infection.** P14-*Nr4a1*-GFP cells were adoptively transferred into mice 1 day prior high dose LCMV clone 13 infection. Animals were sacrificed at various time points. **a** Examples of flow cytometry plots showing expression of *Nr4a1*-GFP and PD-1 in P14 cells (gated on alive single CD8[+] CD45.1[+] cells) isolated from the spleen of naive or chronically infected mice at various days post infection (dpi). Kinetics of GFP frequency **b** and GFP median **c** of P14-*Nr4a1* GFP[+] cells. **d** *Nr4a1*-GFP expression in P14 cells isolated from different tissues 21 dpi compared to the signal observed in activated cells isolated from the spleen 1 dpi. Full circles **b**, **c** and bar plots **d** represent mean ± SD. Empty circles **b**, **c** show individual data points. One of two experiments is shown (*n* = 2 mice for 1 to 3 dpi and 3 mice for 7 to 28 dpi). Each dot represents an individual mouse. See also Supplementary Fig. 1. Source data are provided as a Source Data File.

chronically infected mice, which also expressed high levels PD-1, was significantly lower compared to cells isolated from acutely infected mice. Similar observations were made in virus-specific cells isolated from various tissues such as LN, blood, BM, lung, liver, and kidney (Supplementary Fig. 2). Given the fact that chronic LCMV infection is characterized by high viral titers, the difference between in vivo and in vitro stimulated virus-specific exhausted CD8 T regarding *Nr4a1*-GFP signal cells was surprising.

**Antigen is present 3 weeks into chronic LCMV infection.** The in vitro data suggested that the TCR signaling was still functional to some extent, documented by increased GFP expression after restimulation of exhausted P14-*Nr4a1*-GFP cells. Several hypotheses could explain this difference: (i) the GFP signal is diluted in vivo due to proliferation, (ii) there is little antigen (gp$_{33-41}$/D$^b$) available in vivo despite abundance of infectious viral particles 3 weeks post chronic infection, and (iii) there is strong inhibition of TCR signaling in vivo. Since the proliferation rate of exhausted cells is low in vivo[28], we decided to focus on the last two hypotheses. To test whether antigen sensed by P14 CD8 T cells is present 3 weeks post chronic LCMV infection, naive P14-*Nr4a1*-GFP cells were transferred into 3 weeks chronically infected or naive mice and the GFP signal was assessed 12 h later (Fig. 4a). *Nr4a1*-GFP was highly upregulated in the transferred P14 cells isolated from all screened tissues, suggesting that antigen was abundantly present in mice with established chronic infection, leading to potent activation of naive gp$_{33-41}$-specific CD8 T cells (Fig. 4b, c).

**P14 cells are PD-1[+], while most infected cells are PD-L1[+].** There are numerous reports documenting the role of multiple co-inhibitory receptors on exhausted CD8 T cells, such as CTLA-4, PD-1, LAG3, CD39, TIM-3, 2B4[3,5–8,29–31]. Terminally exhausted cells were shown to co-express multiple co-inhibitory receptors[4] that dampen TCR signaling by various mechanisms[9]. PD-1 is perhaps the most important inhibitory receptor because its absence from the onset of chronic LCMV infection leads to death caused by T-cell-related immunopathology[30], while its blocking at later stages reinvigorates exhausted CD8 T cells[3]. One of its ligands, PD-L1, is expressed on most infected cells (denoted by intracellular staining for the viral nucleoprotein using VL4 antibody) which are found in all screened tissues (Supplementary Fig. 3a, b). Expression of several inhibitory receptors (PD-1, CD39, TIM-3) on P14 cells was high (~50%) during chronic infection (Supplementary Fig. 3c, d); notably, PD-1 is expressed in over 90% of P14 cells isolated from various tissues. Moreover, the fraction of infected hematopoietic cells expressing PD-L1 was quite similar across tissues (Fig. 4a, c), whereas the fraction of PD-L1[+] infected non-hematopoietic cells was higher in peripheral tissues compared to hematopoietic organs (Fig. 5).

**PD-L1 blockade leads to increased TCR signaling in vivo.** In order to assess the effect of PD-1/PD-L1 interaction on TCR signaling in vivo, we performed PD-L1 blockade[3] in mice that had been chronically infected for 3 weeks. Three hours after administration of the blocking antibody, various tissues were harvested and P14 cells were analyzed (Fig. 6a). Blockade efficiency was confirmed by failure of in vitro counterstaining with

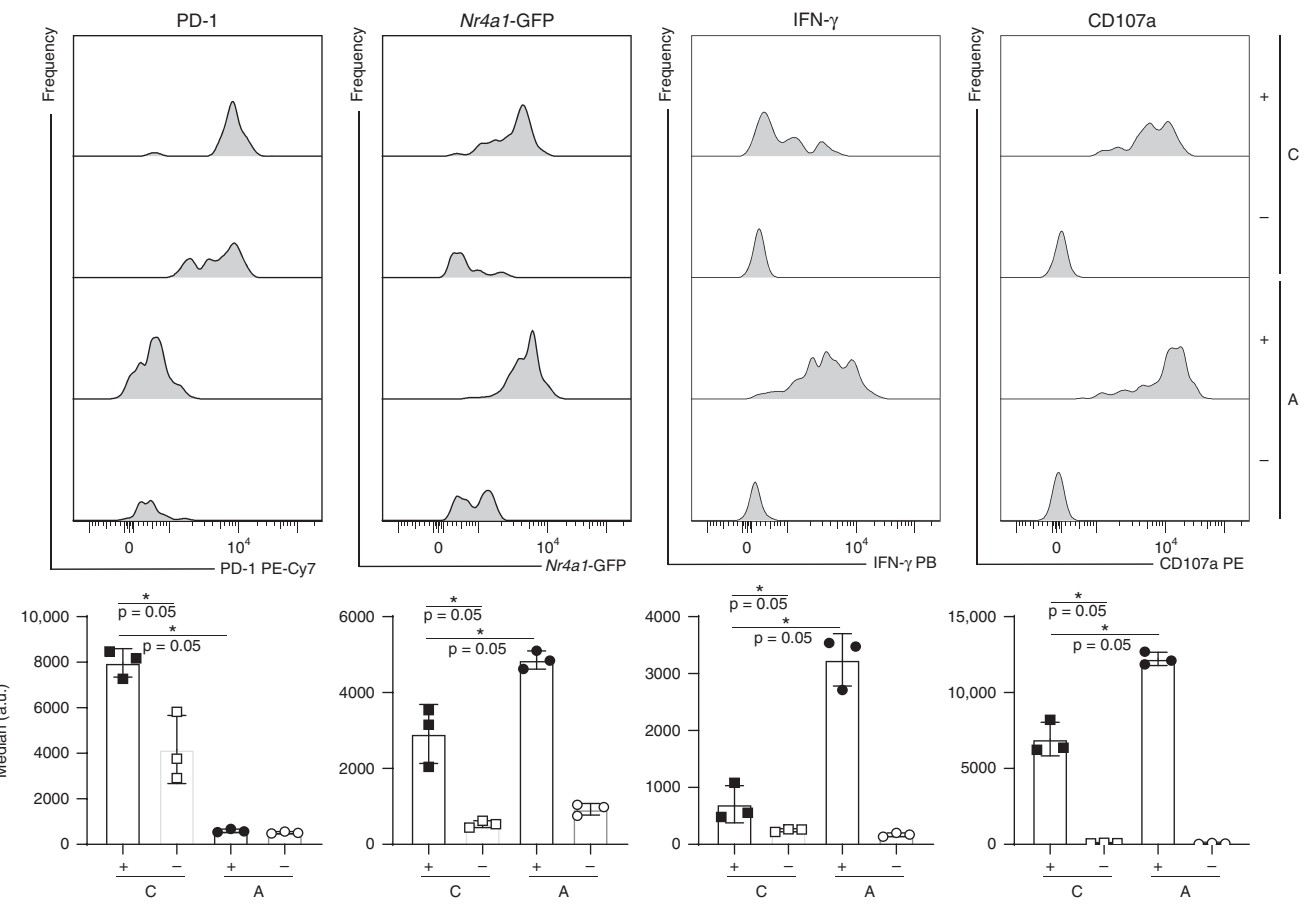

**Fig. 3 In vitro restimulation of sorted P14 cells of acutely or chronically infected mice.** CD45.1+ P14-*Nr4a1*-GFP cells were adoptively transferred into CD45.2+ mice 1 day prior infection with a high (2 × 10⁶ ffu for chronic (C) infection) or low (200 ffu to induce an acute (A) infection) dose of LCMV clone 13. Three weeks post infection, P14 cells were sorted from the spleen and stimulated in vitro for 6 h with plate-bound antibodies. Various markers were quantified (bar plots represent average of medians ± SD) and a representative histogram per group (n = 3 mice) is shown for cells that responded (CD107a+), denoted by plus signs, and cells that did not (CD107a−), denoted by minus signs. Asterisk (*) represents *p*-value = 0.05 (Mann–Whitney one-tailed-test without adjustment for multiple comparisons). Each dot represents an individual mouse. One of two experiments is shown. See also Supplementary Fig. 2. Source data are provided as a Source Data File.

the same clone (Supplementary Fig. 4a). TCR signaling, read out by GFP expression, was significantly increased in the treated group (Fig. 6b–d and Supplementary Fig. 4b, c). Moreover, PD-1 levels were also higher in P14 cells isolated from anti-PD-L1-treated mice compared to the control mice (Fig. 6e and Supplementary Fig. 4d). Since PD-1 is also an activation marker, this increase is likely a result of increased stimulation. Additional short-term co-blockade of other inhibitory receptors (LAG-3, TIM-3, CTLA-4, 2B4, TIGIT) did not significantly enhance TCR signaling compared to PD-L1 blockade (Supplementary Fig. 5), suggesting that PD-1/PD-L1 interaction is the main inhibitory signal.

**In vivo killing of naive targets enhances TCR signaling.** Previous work showed that, despite being less functional than virus-specific cells generated upon an acute infection, exhausted CD8 T cells are capable of efficient in vivo killing of target cells isolated from a naive host[32]. To corroborate that gp₃₃₋₄₁-pulsed target cells originating from naive mice and transferred into recipients with established chronic LCMV infection leads to their killing, and test whether such recognition and killing would be associated with in vivo TCR signaling in exhausted CD8 T cells, peptide-pulsed or unpulsed naive splenocytes were transferred into chronically infected hosts 21 days post infection (Fig. 7a). Gp₃₃₋₄₁-pulsed targets were killed very efficiently only 3 h after

transfer (Fig. 7b, c and Supplementary Fig. 6a). Moreover, TCR signaling was increased in cells isolated from the recipients that received the pulsed targets compared to the control group, which received unpulsed splenocytes (Fig. 7d–f and Supplementary Fig. 6b, c). Given the fact that the virus is not cleared in vivo, the killing efficiency of target cells originating from naive mice might seem surprising. However, there are several factors, which might explain these observations. First of all, the splenocyte suspension is mainly composed of lymphocytes, which are not normally infected by LCMV. Second, antigen expression levels might be different in vivo. Third, likely of highest relevance, the pulsed targets did not express high levels of PD-L1, one of the two ligands that can bind PD-1 in vivo. Indeed, PD-L1 expression on infected cells (denoted by VL4 staining) was several fold higher compared to basal expression on naive splenocytes (Fig. 7g). Hence this last explanation seems the most likely one and would indicate remarkably potent negative regulation of TCR signaling in vivo in exhausted CD8 T cells during established chronic infection. The pool of exhausted cells is very heterogeneous and includes at least three subsets characterized by different phenotypic and functional properties: memory-like expressing TCF1[21,22], CX3CR1hi effector-like and terminally exhausted PD-1hiCXCR6hi cells[18–20]. In vitro killing assays performed with sorted P14 cells and pulsed targets in presence or absence of blocking PD-L1 antibody confirmed the inhibitory role of

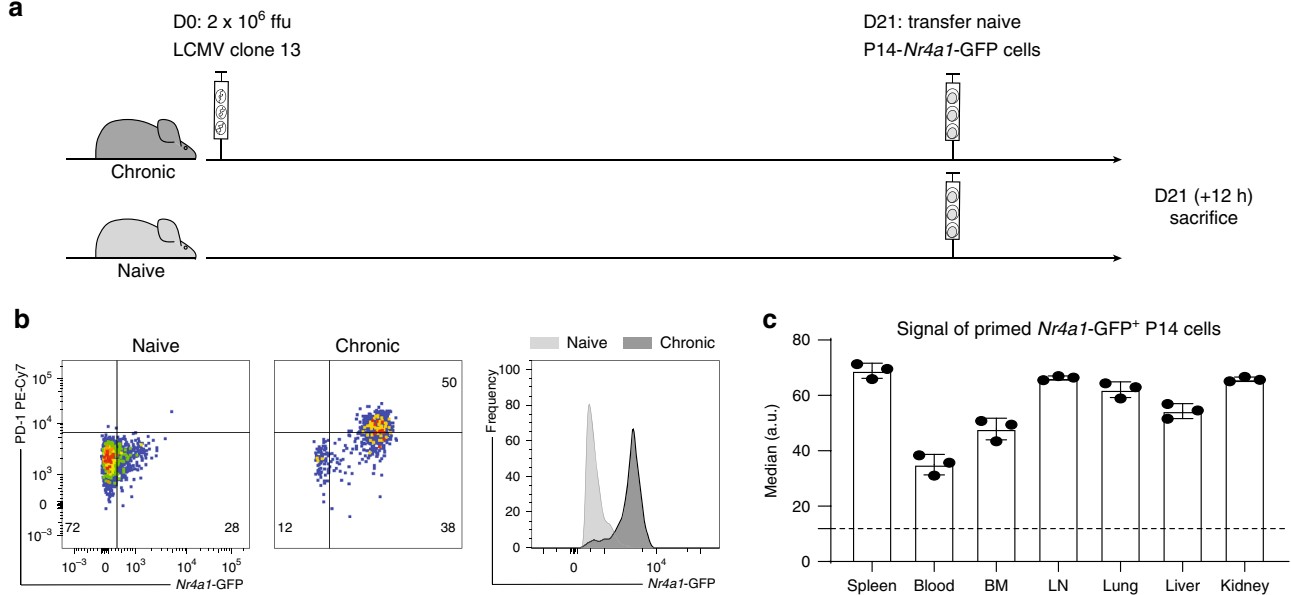

**Fig. 4 Antigen is still present 3 weeks into chronic infection. a** Naive P14-*Nr4a1*-GFP were adoptively transferred into chronically infected or naive hosts. *Nr4a1*-GFP expression in P14 cells (gated on alive single CD8+ CD45.1+ cells) isolated from various tissues was assessed 12 h later. **b** Representative flow cytometry plots and histograms (of P14 isolated from the spleen) show the difference between the two groups (chronically infected in dark gray and naive in light gray). **c** P14 cells isolated from all screened tissues have increased GFP levels (gated on alive single CD8+ CD45.1+GFP+ cells) compared to the signal observed in naive hosts (dotted line). Bar plots show average medians (mean ± SD). Each dot represents an individual mouse. One of three experiments is shown (n = 3 mice). Source data are provided as a Source Data File.

PD-1/PD-L1 axis with respect to cytotoxicity, irrespective of the exhausted subset (Supplementary Fig. 7).

## Discussion

We have characterized TCR signaling in virus-specific exhausted CD8 T cells on the transcriptional level and in an in vivo setting by using the *Nr4a1*-GFP reporter mouse in the context of chronic LCMV infection. The transcriptomic data shows that there are significant differences in virus-specific CD8 T cells from either acutely or chronically infected mice ex vivo without stimulation. These include many co-inhibitory receptors and the transcription factor *Tox*, which has been recently shown to be specifically expressed in exhausted cells[33–35]. However, there are many genes upregulated after in vitro restimulation in both conditions with hardly any difference between the chronic and acute infection, indicating that even CD8 T cells from chronic infection are capable of transcriptionally responding to stimulation.

The low-TCR signaling observed on the transcriptional level was confirmed in vivo, by measuring the activity of the *Nr4a1* promoter used as a proxy for TCR signaling[24]. There was a strong signal induced after initial priming, which was rapidly down-regulated in vivo. The fast decrease of the signal could be attributed, at least at this stage (1–5 days) post infection, to signal dilution due to proliferation and/or downregulation of *Nr4a1*-GFP. Nevertheless, in vitro data showed that the signal is rapidly downregulated before first cell division, suggesting a tight regulation of *Nr4a1*, consistent with its short half-life[25].

Surprisingly, despite abundant antigen being present 3 weeks post infection, TCR signaling is very low in vivo. At this later stage in chronic infection, the low signal is unlikely to be due to proliferation-related dilution because exhausted CD8 T cells do not divide at a fast rate[36]. TCR signaling could be increased by in vitro stimulation of exhausted cells, but not to the same extent as observed in functional CD8 T cells (generated upon an acute infection). Altogether with the RNAseq data and previous studies[2], this suggests that TCR signaling is strongly regulated in

exhausted cells, as maximum signaling levels are not reached in vivo, despite the antigen being abundantly present. Additionally, recent work showed *Nr4a1* transcription is not induced by NFAT alone[37] and there is evidence for ERK signaling mediated AP-1 induction being involved in *Nr4a1* transcription[38]. In chronic LCMV infection, the formation of NFAT/AP-1 dimers is impaired[39], implying that *Nr4a1* does not report the full extent of TCR signaling in this setting.

IFN-γ secretion and degranulation were also significantly lower in exhausted cells compared to functional cells (generated upon acute LCMV infection), as previously shown[28,40] (Fig. 3 and Supplementary Fig. 3). Not surprisingly, exhausted virus-specific CD8 T cells co-expressed a multitude of inhibitory receptors, which dampen TCR signaling[4]. Indeed, both short-term PD-L1 blockade and adoptive transfer of pulsed target cells isolated from naive mice led to increased *Nr4a1*-GFP and PD-1 expression in vivo, linked to efficient killing of these target cells as shown previously[41]. The highest increase in TCR signaling was observed in P14-*Nr4a1* cells isolated from spleen and lungs after adoptive transfer of pulsed target cells isolated from naive mice, probably due to the nature and delivery of targets. The pulsed cells were splenocytes, mainly composed of naive lymphocytes, which are primarily in circulation and home to secondary lymphoid tissues. Additionally, due to the intravenous delivery, most targets would initially reach the lungs where there are many P14 cells[30] that could kill the pulsed targets specifically, resulting in fewer pulsed targets reaching other peripheral organs. Importantly, the adoptively transferred target cells from naive mice expressed lower levels of PD-L1 compared to VL4+ LCMV-infected cells in chronically infected hosts, thus, lowering negative regulation of TCR signaling in exhausted CD8 T cells. This difference might explain why naive targets are recognized and eliminated, while most endogenous infected targets are not[42]. Altogether, these results suggest that TCR signaling is strongly inhibited in vivo.

Compared to PD-L1 blockade alone, short-term co-blockade of several inhibitory receptors (PD-1, LAG-3, CTLA-4, TIM-3,

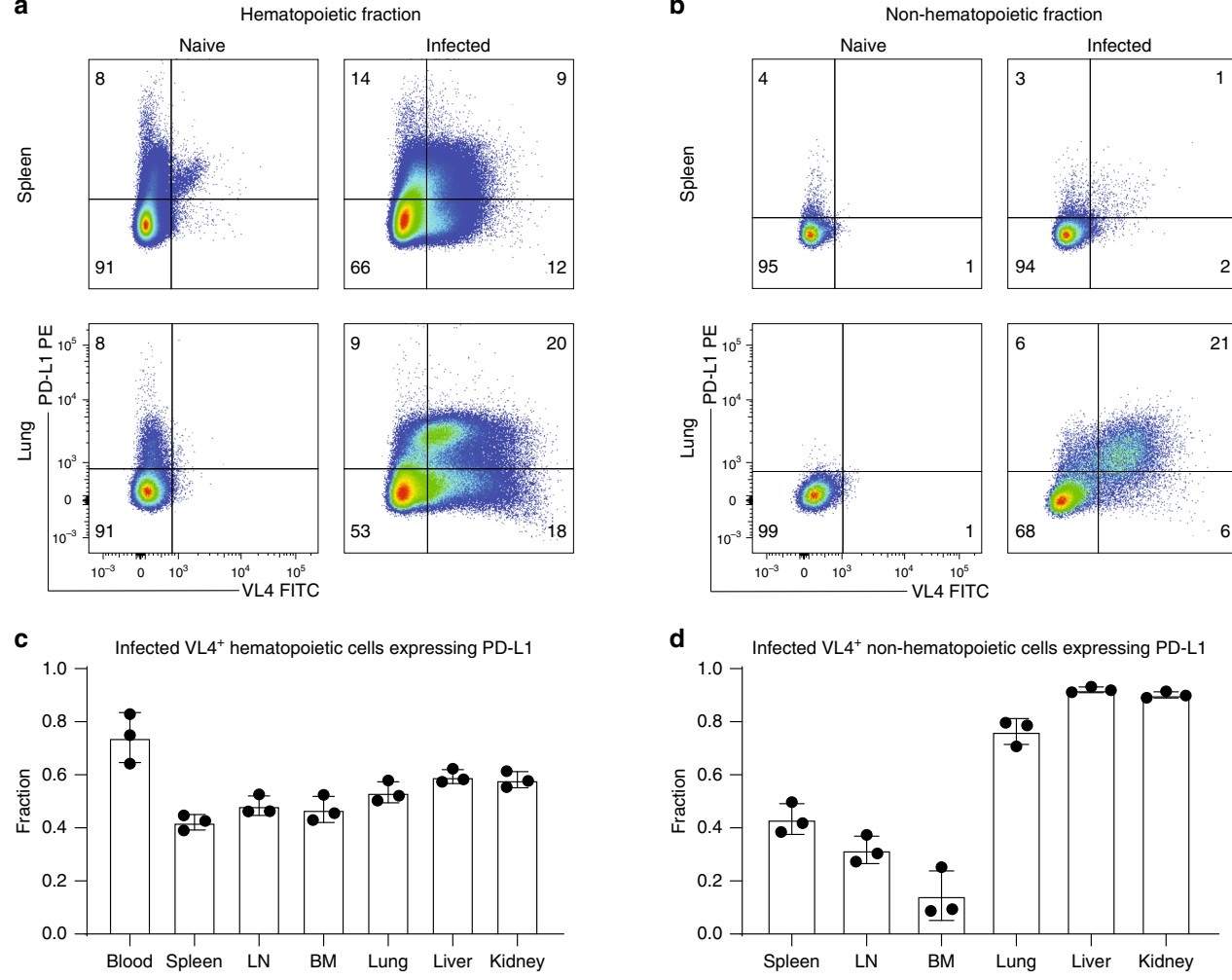

**Fig. 5 PD-L1 expression on infected cells isolated 3 weeks into chronic infection.** Seven tissues from chronically LCMV-infected mice were analyzed for infected (VL4$^+$) cells 21 days post infection. Representative flow cytometry plots show PD-L1 and VL4 expression in hematopoietic (spleen) and peripheral tissues (lung) in the hematopoietic fraction **a** (gated on alive CD45$^+$CD45.2$^+$ cells) or in the non-hematopoietic fraction **b** (gated on alive CD45$^-$CD45.2$^-$ cells). Fraction of infected hematopoietic **c** or non-hematopoietic **d** cells (gated on VL4$^+$) expressing PD-L1 is quantified in the bar plots showing mean ± SD. One of two experiments is shown (n = 3 mice). Each dot represents an individual mouse. See also Supplementary Fig. 3. Source data are provided as a Source Data File.

TIGIT) did not show a significant increase of *Nr4a1*-GFP expression in exhausted LCMV-specific CD8 T cells in vivo (Supplementary Fig. 5), suggesting that the PD-1/PD-L1 axis exerts the strongest inhibition on TCR signaling. This observation is consistent with previous work, which showed that PD-1/PD-L1 interaction protects mice from CD8-mediated lethal immuno-pathology[30]. Several studies proved that co-blockade of PD-L1 and another inhibitory receptor had synergistic effects, showing benefits over PD-L1 blockade alone[9]. Moreover, in some cases, blocking an inhibitory receptor is only efficient when combined with PD-L1 blockade. For example, blocking LAG-3 in a chronic LCMV infection has no effect on the function of exhausted virus-specific CD8 T cells[6], while co-blockade of PD-L1 and LAG-3 leads to their reinvigoration[4]. This suggests that blocking LAG-3 is only efficient when PD-1/PD-L1 interaction is inhibited. The fact that there was no increase of TCR signaling compared to single PD-L1 short-term blockade suggests that the direct inhibition of TCR signaling transduction is mainly due to the PD-1/PD-L1 axis. Nevertheless, other inhibitory receptors are able to interfere with signaling cascade downstream the TCR as well. For example, CTLA-4 can inhibit AKT[11], while TIGIT ligation can lead to downregulation of TCR components[43]. However, unlike

PD-1 whose ligand is widely expressed[44], the ligand expression of CTLA-4 and TIGIT is restricted to APCs[10,11]. It is possible that the short-term blockade did not allow for a substantial fraction of exhausted cells to interact with target cells expressing ligands of these inhibitory receptors. Our results suggest that, out of the screened blocking antibodies (PD-L1, LAG-3, CTLA-4, TIM-3, TIGIT), PD-1 is the main inhibitory receptor dampening TCR signaling. Once this axis is hampered, blocking other receptors that have complimentary inhibitory mechanisms (such as effects on proliferation, cytokine production or interference with co-stimulatory pathways[9]) could lead to a synergistic effect.

In summary, exertion of effector functions of exhausted CD8 T cells is strongly and actively inhibited in vivo by potent inhibition of TCR signaling via engagement of co-inhibitory receptors. In their absence, exhausted CD8 T cells are able to resume function and, above all, their cytotoxic potential.

## Methods
**Bulk RNAseq**. P14 cells were sorted (BD FACS Aria, BD Biosciences) from animals with either acute or chronic LCMV infection. After 4 h of in vitro restimulation in the presence of anti-CD3 (clone 145-2C11, Biolegend) and anti-CD28 (clone 37.51, Biolegend) antibodies, RNA was extracted using TRIzol

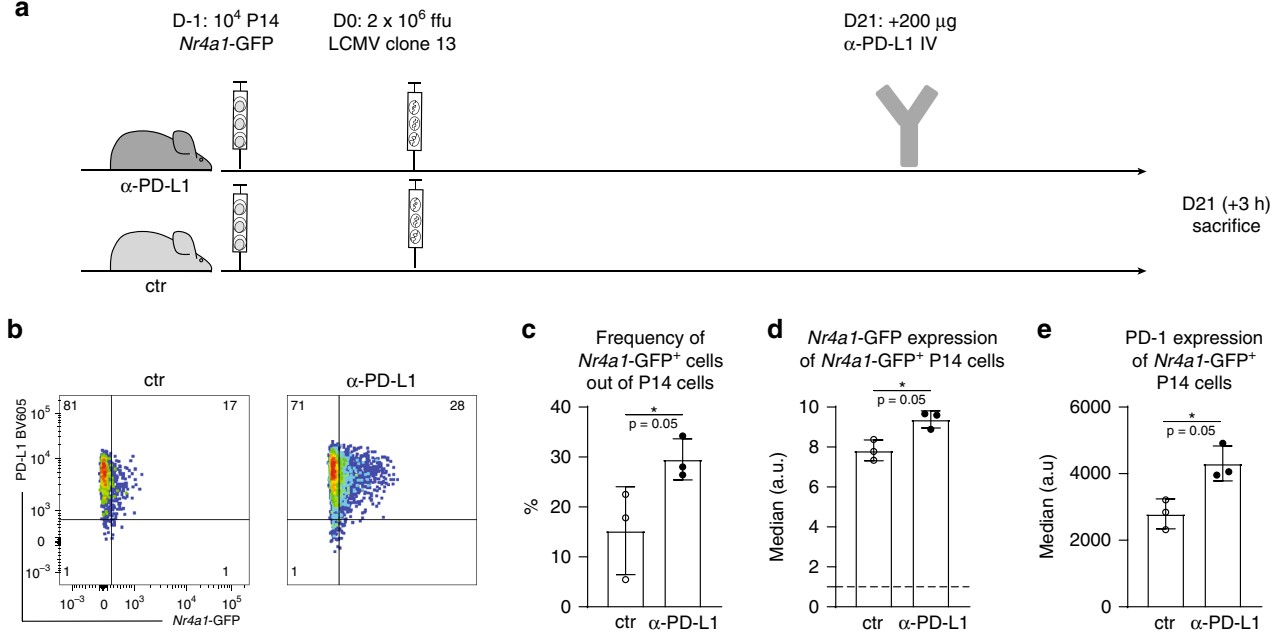

**Fig. 6 Increase in TCR signaling after short-term PD-L1 blockade. a** P14-*Nr4a1*-GFP were adoptively transferred into naive hosts 1 day prior high-dose LCMV clone 13 infection. Twenty-one days post infection, one group received one dose of anti-PD-L1 blocking antibody intravenously. The mice were sacrificed 3 h after treatment. **b** Representative plots showing *Nr4a1*-GFP signal in two groups, control (ctr) and treated (α-PD-L1) are shown. Fraction of *Nr4a1*-GFP + cells (gated on alive single CD8$^+$ CD45.1$^+$ P14 cells) (**c**)), median of GFP (**d**) and PD-1 (**e**) (gated on alive single CD8$^+$ CD45.1$^+$ CD45.1 GFP$^+$ cells) are shown for cells isolated from the spleen. Bar plots represent mean ± SD. Asterisk (*) represents *p*-value = 0.05 (Mann–Whitney one-tailed-test). One of three experiments is shown (*n* = 3 mice). Each dot represents an individual mouse. See also Supplementary Figs. 4 and 5. Source data are provided as a Source Data File.

(Thermofisher Scientific) according to the manufacturer's protocol and processed for sequencing at the Functional Genomics Center Zürich (FGCZ). The extracted RNA was selected for messenger RNA (mRNA) transcripts using polyA-tail capture. The enriched mRNA was reverse transcribed and sequenced on the Illumnia HiSeq 4000 platform. Resulting transcript counts were analyzed and differential gene expression was analyzed using the DEseq2 package[45] in R[46].

**Mice.** Wild-type male CD45.2 C57BL/6J mice purchased from Janvier Elevage, *Nr4a1*-GFP mice expressing GFP under the control of the NUR77 promoter[24], *Tcf7*-GFP mice expressing GFP under the control of *Tcf7* (encoding TCF1) promoter[21], P14 transgenic (CD45.1) mice expressing a TCR specific for LCMV peptide gp$_{33-41}$[47] were housed at 24 °C and 50% humidity and bred under specific pathogen-free conditions at the ETH Phenomics Center Hönggerberg. Mice were exposed to a 12:12 h light–dark cycle with unrestricted access to water and food. All mice used in experiments had between 6 and 16 weeks. P14-*Nr4a1*-GFP or P14-*Tcf7*-GFP were generated by crossing *Nr4a1*-GFP or *Tcf7*-GFP mice to P14 mice. All animal experiments were conducted according to the Swiss federal regulations and were approved by the Cantonal Veterinary Office of Zürich (Animal experimentation permissions 147/2014, 115/2017).

**Virus.** LCMV clone 13 was propagated on baby hamster kidney 21 cells (BHK21 [C13] (ATCC® CCL10). Viral titers of virus stocks were determined as described previously by using MC57G (ATCC® CRL-2295) cells[48].

**Infections.** In all, 10$^4$ transgenic cells (P14 or P14-*Nr4a1*-GFP) were adoptively transferred 1 day prior LCMV clone 13 intravenous (IV) infection with $2 \times 10^6$ or 200 ffu/mouse to induce a chronic or acute infection, respectively. For analysis at 1, 2, and 3 days post infection, $5 \times 10^5$ P14-*Nr4a1*-GFP cells were transferred.

**Cell isolation from tissues.** Mice were sacrificed with carbon dioxide. Blood was taken from the heart or vena cava. The mice were then perfused with 20 ml PBS, and organs (spleen, bone marrow, lymph nodes, lung, liver, and kidney) were isolated. Lungs, liver, and kidney were cut into pieces and digested in complete RPMI (Bioconcept) (RPMI-1640 containing 10% fetal bovine serum (Omnilab), 2 mM L-glutamine (Gibco), 1% penicillin–streptomycin (Gibco), 1 mM sodium pyruvate (Gibco), 50 nM betamercapthoethanol, 0.1 mM non-essential (glycine, L-alanine, L-asparagine, L-aspartic acid, L-glutamic acid, L-proline, L-serine) amino acids (Gibco), 20 mM HEPES (Gibco)) supplemented with 2.4 mg/ml collagenase type I (Gibco), and 0.2 mg/ml DNase I (Roche Diagnostics, Rotkreuz, Switzerland) for 30 min at 37 °C. Bone marrow was flushed with a syringe containing 5 ml of

complete RPMI. Spleens and lymph nodes were mashed through 70 μm filter with a syringe (1 ml) plunger. Cell suspension from lungs, liver, and kidney were enriched for lymphocytes by centrifugation over Percoll density centrifugation (30% (v/v) Percoll in PBS) at 4 °C (500 × *g*) for 30 min. Cell suspensions were filtered (70 μm) and treated with ammonium-chloride-potassium buffer (150 mM NH$_4$Cl, 10 nM KHCO$_3$, 0.1 mM EDTA in water) to lyse erythrocytes for 5 min at room temperature.

**Cell sorting.** Spleen samples were depleted of CD4 and B cells by incubating splenocyte suspensions in enrichment buffer (PBS, 1% FCS, 2 mM EDTA) with biotinylated α-CD4 and α-B220 antibodies (1/100 dilution) at room temperature for 20 min, followed by incubation with streptavidin-conjugated beads (Mojo, Biolegend) (4%) for 5 min at room temperature. Cells were then placed on a magnetic separator (StemCell) for 3 min at room temperature, followed by collection of supernatant. Enriched samples from the spleen or cell suspensions from other tissues of interest (see "Cell isolation from tissues") were stained with α-CD8 and α-CD45.1 to sort P14 cells (ARIA cell sorter, BD Biosciences).

**In vitro restimulations.** Restimulations of sorted virus-specific P14-*Nr4a1*-GFP cells were performed in 96-well flat-bottom plates pre-coated with anti-CD3 and anti-CD28 at 1 μg/ml. Sorted cells were incubated in complete RPMI at 37 °C for 6 h at 37 °C in presence of 2 μM monensin A, 10 μg/ml brefeldin, and α-CD107a antibody.

**Flow cytometric analysis.** Surface staining was performed at room temperature for 30 min in FACS buffer (2% FCS, 1% EDTA in PBS). LIVE/DEAD™ Fixable Near-IR Dead (Thermo Fisher) was used to discriminate alive from dead cells. Fluorophore-conjugated antibodies used for flow cytometry were purchased from eBioscience (San Diego, California, USA) (α-IFNγ PE XMG1.2), and BioLegend (Lucerna Chem AG, Luzern, Switzerland) (α-IFNγ Pacific Blue XMG1.2, α-CD107a PE 1D4R; α-CD44 BV510 IM7; α-CD45.1 Pacific Blue A20; α-CD45.1 APC A20; α-CD8 PerCP 53-6.7; α-CD39 AF647 A7R34; α-TIM-3 PE RMT3-23; α-PD-1 PE-Cy7 29F.1A12, α-PD-L1 PE 10F.9G2, α-CD45.2 Pacific Blue 104, α-CXCR6 PE SA051D1, α-CX3CR1 PE-Cy5.5 SA011F11), BD Biosciences (Allschwil, Switzerland) (α-CD45 PerCP 30-F11) or self-produced (VL4-FITC in VL4 hybridoma). Intracellular staining was performed according to the manufacturer's protocol (Biolegend (intracellular cytokine staining kit). Most antibodies were used at a 1/200 dilution except for IFN-γ and VL4 (1/50), and CD107a (1/1000). Data was acquired LSR II and LSR II Fortessa using Diva software (BD Biosciences, Allschwil, Switzerland) and analyzed in FlowJo 10.6.1 and 9 (BD Biosciences, Allschwil, Switzerland). Plots were generated and statistical analysis

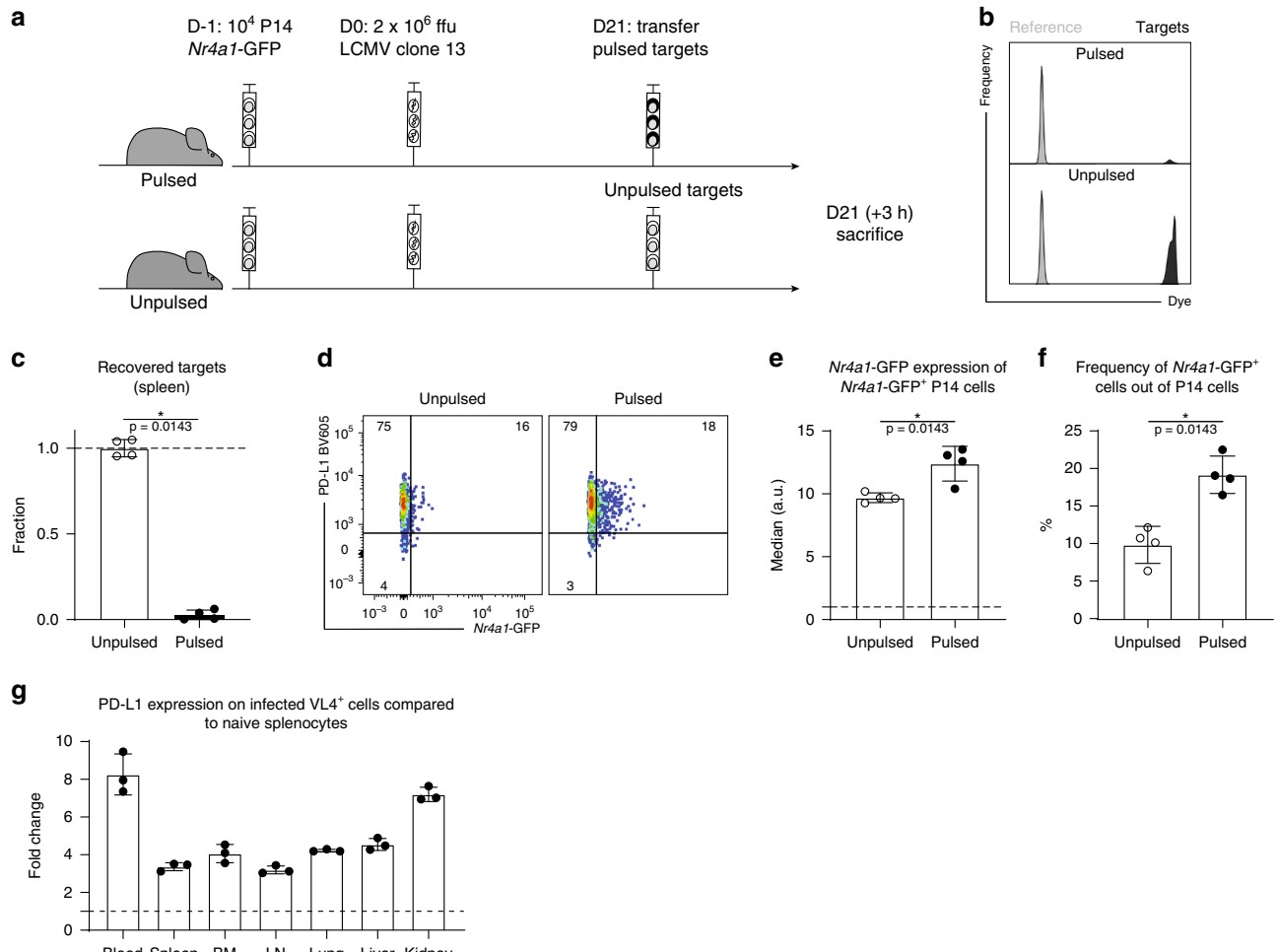

**Fig. 7 In vivo cytotoxicity of exhausted CD8+ T cells 3 weeks into chronic LCMV infection. a** CD45.1+ P14-*Nr4a1*-GFP cells were adoptively transferred into naive CD45.1+ mice 1 day prior to high-dose LCMV clone 13 infection. Twenty-one days post infection, the hosts received gp$_{33-41}$-pulsed or unpulsed labeled targets together with a reference population. Three hours later the mice were sacrificed. **b** Representative plots of transferred targets isolated from the spleen. **c** Fraction of recovered targets (calculated based on the reference population) in spleens isolated from unpulsed or pulsed groups is shown. **d** Representative flow cytometry plots showing ex vivo *Nr4a1*-GFP expression in P14 cells (gated on alive single CD8+ CD45.1+ cells) from the spleens of mice, which received unpulsed or pulsed targets. Medians of GFP (gated on alive single CD8+ CD45.1+ GFP+ cells) (**e**)) and frequency (**f**) of GFP+ cells (out of P14 cells) are shown. **g** Fold change of PD-L1 expression between infected cells (gated on alive single VL4+ cells) isolated from different tissues 3 weeks into chronic LCMV infection and naive splenocytes (dotted line). Bar plots represent mean ± SD. Asterisk (*) denotes a significant *p*-value ≤ 0.05 (Mann–Whitney one-tailed-test). One of three experiments is shown (*n* = 4 (**c**–**f**), 3 (**g**) mice). Each dot represents an individual mouse. See also Supplementary Fig. 6. Source data are provided as a Source Data File.

was performed in GraphPad Prism 8.2.0 (La Jolla, California, USA). Gating strategy is shown in Supplementary Fig. 7. In order to facilitate comparison of GFP medians between different experimental time points, GFP values were normalized by the values of a reference population (GFP median of non transgenic CD8+ cells). GFP values shown in Figs. 2c, 4c, 6d, 7e and Supplementary Figs. 1d, 5d, 6c are normalized and represent fold increase compared to the reference population.

**In vivo blocking**. One dose of blocking antibodies (200 µg of α-PD-L1 (clone 10 F.9G2), α-TIM-3 (clone RMT3-23), α-LAG-3 (clone C9B7W), α-CTLA-4 (clone UC10-4F10-11), α-TIGIT (clone 1B4)[49]) were administered IV 3, 6, or 12 h prior sacrifice.

**In vivo killing assay**. Splenocytes were incubated at 37 °C for 1 h with or without 0.1 µg/ml gp$_{33-41}$ in RPMI, labeled with cell proliferation dye (eFluor 450 or Cell Trace Yellow, Thermo Fisher) according to manufacturer's protocols. The reference population and the target population (pulsed or unpulsed) were mixed (15 × 10[6] splenocytes per population) and transferred IV into chronically infected mice. After 3 h, the animals were sacrificed and target cell abundance was quantified by flow cytometry.

**In vitro killing assay**. CD45.1+ P14-*Tcf7*-GFP (transgenic P14 cells expressing GFP under the control of the TCF1 promoter, encoded by *Tcf7*) cells were

transferred into CD45.2+ hosts 1 day prior chronic LCMV clone 13 infection. P14 cells were sorted from organs (lungs, liver, and spleen) of chronically infected mice 21 dpi based on CX3CR1, CXCR6, and TCF1-GFP into different subsets. Target cells (EL4 cells) were incubated with IFN-γ at 100 U/ml (PeproTech) for 6 h prior pulsing to induce PD-L1 upregulation and pulsed with gp$_{33-41}$ peptide at 37 °C for 1 h, then washed with medium. For one condition, PD-L1 was blocked with 30 µg/ml α-PD-L1 (clone 10F.9G2) for 1 h hour at 37 °C. Sorted populations were incubated with unpulsed or pulsed (in the presence or absence of α-PD-L1). EL4 target cells at 5:1 *E/T* ratio. Counting beads (CaliBRITE, BD Biosciences) were added to the samples stained for flow cytometry.

**Statistical analysis**. Graphpad prism 8.2.0 software or R was used to calculate significance between the samples. *p*-values ≤ 0.05 were considered significant. Statistical test is indicated in each figure.

**Reporting summary**. Further information on research design is available in the Nature Research Reporting Summary linked to this article.

## Data availability
RNA sequence data are deposited in the European Nucleotide Archive under accession code PRJEB38896. The flow cytometry files supporting the findings are available from the corresponding author upon request. Source data are provided with this paper.

## Code availability

Custom R script used to analyze the RNA sequencing data is available here: https://gitlab.com/claassen_lab/tcrsignaling.

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

## Acknowledgements

We thank Prof. Roman Spörri for providing *Nr4a1*-GFP reporter mice[24], Prof. Werner Held for providing *Tcf7*-GFP reporter mice[21], and Prof. Nicole Joller for providing the TIGIT blocking antibody. We would like to thank Nathalie Oetiker and Franziska Wagen for excellent technical assistance. We are grateful to the members of the Claassen, Oxenius, Joller (University of Zurich), and Sallusto groups for helpful discussions and feedback. Funding: This work was supported by the ETH Zurich (Grant No. ETH-39 14-2 to MC and AO).

## Author contributions

I.S., D.C., M.C., A.O. designed the experiments; I.S., D.C. carried out the experiments, I.S., D.C., M.C., A.O. analyzed the experiments; I.S., D.C., M.C., A.O. wrote the manuscript.

## Competing interests

The authors declare no competing interests.
