## [Peer Review File · Nature Communications]

REVIEWER COMMENTS

Reviewer #1 (Remarks to the Author):

In the manuscript titled "T cell receptor signaling in exhausted virus specific CD8 T cells is low and strongly inhibited in vivo during chronic LCMV infection" the authors investigated the level of TCR signaling that virus-specific T cells experience during chronic LCMV infection. Using the LCMV model of acute and chronic viral infection, the authors initially performed RNAseq gene expression profiling on functional and exhausted antigen-specific T cells to assess the baseline differences in gene expression as well as the changes that occur during a 4 hour ex vivo stimulation. The authors observed several genes downstream of TCR signaling that were refractory to upregulation during ex vivo stimulation. The authors proceeded to investigate the potential in vivo suppression of TCR signaling by analyzing Nur77 expression (GFP reporter) in P14 transgenic T cells during chronic LCMV infection. Despite having an abundance of antigen, the authors note that the reporter signaling declines significantly after the peak of the effector response (~ 7 days post chronic LCMV infection). The authors next demonstrated that naive P14 cells (with Nur77 reporter) exhibit high TCR signaling when adoptively transferred into chronically infected mice, indicating that the chronically stimulated likely have access to antigen. Thus, the authors conclude that the suppression of TCR signaling is due to an adaptation to the chronically stimulated T cells that limits their ability to be activated. Having demonstrated that chronically stimulated T cells have an inhibitory signal that limits T cell signaling, the authors proceeded to investigate the impact of PD-1-PDL1 ligation in limiting TCR signaling. To examine this mechanism the authors first performed PDL1 blockade experiments using the Nur77 P14 reporter cells and found a modest increase in Nur77 expression after checkpoint blockade. Extending this observation, the authors next transferred peptide-pulsed target cells into chronically infected mice and reported significant antigen-dependent cytotoxicity. Overall, the experimental approach and data are presented very clearly, however the results appear to be largely confirmatory of many prior observations. Additionally, the study overlooks the heterogeneity that exists among the pool of "exhausted" P14 cells. For instance, several groups have reported stem versus terminally exhausted T cells among the total pool of chronically stimulated LCMV-specific CD8 T cells. Do the authors know if the minor subset of stem-like P14 cells provide the cytotoxicity shown in figure 7 or is the cytotoxicity derived from all cells equally? The answer to this question has significant implications for how the data are interpreted.

Specific comments:

- 1) Generally, there is very little functional and phenotypic data provided to support the claim that the P14 cells from the chronically infected animals are indeed exhausted. Additionally, many of the experiments are performed at the two-week time point which is not typical for characterizing exhausted T cells in the LCMV model system. The authors should provide viral titers from these time points.
- 2) Given that TCR signaling is often associated with a calcium flux, it was surprising that such measurements were not described.
- 3) The authors describe VL4 staining as a readout for virally infected cells, but do not confirm that the cells indeed contain infectious viral particles. While this is very likely correct, confirmation of this assay by direct measurement of virus should be performed.

Reviewer #2 (Remarks to the Author):

CD45.1+ P14 CD8 T cells specific for the LCMV-derived gp33-41 peptide were adoptively transferred into CD45.2+ hosts one day prior infection with a low (200 ffu) or high (2x10⁶ ffu) dose of LCMV clone 13 to induce acute and chronic infections, respectively. Two weeks later, P14 cells were isolated from the spleens and restimulated in vitro with plate-bound α -CD3 and α -CD28 for 4 h and their transcriptional profile defined using bulk RNAseq. Cells isolated from chronically

infected mice have a distinct transcriptional profile compared to the ones isolated from acutely infected mice. Upon in vitro stimulation, 429 genes showed similar changes in expression in both CD8 T cells isolated from acutely and chronically infected mice. Only 35 genes were differentially expressed in exhausted cells when compared to effector/memory cells after stimulation. CD45.1+ P14-Nr4a1-GFP CD8 T cells were adoptively transferred into wild type CD45.2+ C57BL/6J mice one day prior infection with a high dose of 2×10^6 ffu LCMV clone 13. One day after chronic infection the Nr4a1-GFP reporter and PD-1 inhibitory receptor were upregulated and then rapidly downregulated in cells isolated from spleen and various infected tissues. P14-Nr4a1-GFP cells isolated three weeks after chronic or acute LCMV infection and stimulated in vitro with anti-CD3 and anti-CD28 antibodies for 6 hours upregulated the Nr4a1-GFP reporter. However, Nr4a1-GFP signals from cells isolated from chronically infected mice were lower than those of T cells isolated from acutely infected mice. Antigen was present in CD45.2+ C57BL/6J recipients three weeks after chronic LCMV infection and can not thus account for the lack of TCR signaling in vivo (at least using the Nr4a1-GFP reporter). Considering that PD-1 is expressed on exhausted P14-Nr4a1-GFP T cell and that PDI-L1 is expressed on infected cells three weeks after chronic LCMV infection, α -PD-L1 was administered to those mice and P14-Nr4a1-GFP cells isolated from various tissues were analyzed three hours after administration. approximately a quarter of P14-Nr4a1-GFP cells expressed some weak levels Nr4a1-GFP signals. Addition of antibodies against other inhibitory receptors (LAG-3, TIM-3, CTLA4, 2B4, TIGIT) did not increase the Nr4a1-GFP signals or the frequency of responding cells. Finally, gp33-41-pulsed target cells originating from naïve mice were transferred into recipients with established chronic LCMV infection and it was analyzed whether their killing would be associated with in vivo TCR signaling in exhausted P14-Nr4a1-GFP T cells. As compared to control mice that received unpulsed target cells, 10% P14-Nr4a1-GFP cells expressed Nr4a1-GFP signals. The elegant and novel data presented in this paper are fairly discussed and in particular the observation that among the screened blocking antibodies PD-1 is the most efficient to rescue Nr4a1-GFP signals. However, most of the data are shown using histograms with % and MFI and as emphasized in Specific Comments it prevents proper evaluation of the data.

Specific issues.

1/ The paper is sometimes difficult to read and some words are likely missing in the following sentences : 'Previous studies showed that virus-specific CD8 T cells are transcriptionally distinct when compared to virus-specific CD8 T cells isolated upon acute infection and fail to exert several effector functions such as production of the inflammatory cytokines IFN- γ and TNF- α .' And 'TCR transgenic CD45.1+ P14 CD8 T cells that recognize the LCMV-derived gp33-41 peptide were adoptively transferred into CD45.2+ hosts one day prior infection with a low (200 ffu) or high (2×10^6 ffu) dose of LCMV clone 13 to induce an acute, respectively chronic infection.'

2/ Work by the group of Hogquist (PMID: 25137456) shows that T cells express basal /homeostatic levels of Nr4a1-EGFP reporter. Accordingly, a negative control should be shown in Fig. S1 corresponding to P14 CD8 T cells lacking the Nr4a1-EGFP reporter.

3/ Fig. 2. A Nr4a1-GFP vs PD-1 dot plot corresponding to CD45.1+ P14-Nr4a1-GFP CD8 T cells from non-infected mice need to be shown.

4/ Fig. 3. Is the % of CD107-positive cells similar in the two experimental conditions ?

5/ Fig. S3C. FACS profiles and not only histograms displaying % expression need to be shown for PD-1, CD39, and TIM-3 at 2 to 4 weeks post-infection.

6/ Fig. 7D. Plots similar to the one shown in 6B need to be shown.

7/ Fig. 7F. FACS profiles need to be shown.

8/ Are P14-Nr4a1-GFP CD8 T cells found in established chronic LCMV infection expressing BTLA ?

Point by point reply "T cell receptor signaling in exhausted virus-specific CD8 T cells is low and strongly inhibited in vivo during chronic LCMV infection"

We would like to thank both reviewers for their critical, constructive and very valuable comments. We believe that we can address all the raised concerns as detailed below.

Specific replies to the individual reviewers:

Reviewer #1 (Remarks to the Author):

In the manuscript titled "T cell receptor signaling in exhausted virus specific CD8 T cells is low and strongly inhibited in vivo during chronic LCMV infection" the authors investigated the level of TCR signaling that virus-specific T cells experience during chronic LCMV infection. Using the LCMV model of acute and chronic viral infection, the authors initially performed RNAseq gene expression profiling on functional and exhausted antigen-specific T cells to assess the baseline differences in gene expression as well as the changes that occur during a 4 hour ex vivo stimulation. The authors observed several genes downstream of TCR signaling that were refractory to upregulation during ex vivo stimulation. The authors proceeded to investigate the potential in vivo suppression of TCR signaling by analyzing Nur77 expression (GFP reporter) in P14 transgenic T cells during chronic LCMV infection. Despite having an abundance of antigen, the authors note that the reporter signaling declines significantly after the peak of the effector response (~ 7 days post chronic LCMV infection). The authors next demonstrated that naïve P14 cells (with Nur77 reporter) exhibit high TCR signaling when adoptively transferred into chronically infected mice, indicating that the chronically stimulated likely have access to antigen. Thus, the authors conclude that the suppression of TCR signaling is due to an adaptation to the chronically stimulated T cells that limits their ability to be activated. Having demonstrated that chronically stimulated T cells have an inhibitory signal that limits T cell signaling, the authors proceeded to investigate the impact of PD-1-PDL1 ligation in limiting TCR signaling. To examine this mechanism the authors first performed PDL1 blockade experiments using the Nur77 P14 reporter cells and found a modest increase in Nur77 expression after checkpoint blockade. Extending this observation, the authors next transferred peptide-pulsed target cells into chronically infected mice and reported significant antigen-dependent cytotoxicity. Overall, the experimental approach and data are presented very clearly, however the results appear to be largely confirmatory of many prior observations. Additionally, the study overlooks the heterogeneity that exists among the pool of "exhausted" P14 cells. For instance, several groups have reported stem versus terminally exhausted T cells among the total pool of chronically stimulated LCMV-specific CD8 T cells. Do the authors know

if the minor subset of stem-like P14 cells provide the cytotoxicity shown in figure 7 or is the cytotoxicity derived from all cells equally? The answer to this question has significant implications for how the data are interpreted.

This study focuses on the differences between acute and chronic LCMV infections. While it's true that there are several subsets of exhausted cells, all of them are completely different from any cells generated in acute infection, so a direct comparison of these subsets is difficult. The focus of the paper was not on different subsets in chronic infection, but on the ability of the ensemble of exhausted cells generated during chronic infection to kill naïve targets. The potentially different cytotoxic activity of these subsets do not change the fact that the pool of exhausted cells is able to eliminate these targets.

To directly respond to the question raised by the reviewer, we have assessed the cytotoxicity of different sorted subsets (CXCR6^{hi} exhausted, CX3CR1^{hi} effector-like or TCF1^{hi} memory-like cells) *in vitro* (Fig 1, below), but we have not observed differences between the different subsets. However, the *in vitro* cytotoxicity assay does not entirely mimic the *in vivo* conditions: the microenvironment regarding inhibitory ligand expression on infected cells, cytokine milieu, or amount of infected cells might be different for different subsets within tissues. Nevertheless, locking PD-L1 ligand *in vitro* also leads to a higher cytotoxicity, further confirming the role of PD-1 on restraining TCR signaling and hence cytotoxic effector functions of exhausted CD8 T cells.

Fig. 1: P14-*Tcf7* (transgenic P14 cells expressing GFP under the control of the TCF1 promoter, encoded by *Tcf7*) cells were sorted from chronically infected mice 21 dpi based on CX3CR1, CXCR6, and TCF1-GFP expression into advanced exhausted CXCR6^{hi}CX3CR1^{neg}, effector-like CX3CR1^{hi}CXCR6^{lo}, and memory-like GFP^{hi} cells. Sorted populations were incubated with GP₃₃₋₄₁ pulsed EL4 target cells at 5:1 E/T ratio. EL4 cells were incubated with IFN-γ at 100 U/mL for 6 hours prior pulsing to induce PD-L1 upregulation (data not shown). For one condition, PD-L1 expression was blocked with 30 μg/mL α-PD-L1 for one hour at 37°C (“α-PD-L1” denotes the condition with pulsed targets were incubated with α-PD-L1, while “ctr” indicates the condition with pulsed targets). The graph shows percentage of killing; lines denote paired samples. Data is from one experiment n=1-3.

Specific comments:

1) Generally, there is very little functional and phenotypic data provided to support the claim that the P14 cells from the chronically infected animals are indeed exhausted. Additionally, many of the experiments are performed at the two-week time point which is not typical for characterizing exhausted T cells in the LCMV model system. The authors should provide viral titers from these time points.

The RNAseq data was generated with cells isolated at 14 dpi because it was easier to recover sufficient P14 numbers from chronically infected mice for all experimental conditions from one mouse only (without the need to pool several mice in order to have enough cells for all experimental conditions). Previous studies (Wherry et al., 2007) showed that virus-specific CD8 T cells show signs of exhaustion already two weeks into chronic infection.

Editorial Note: Figure 2 in this Peer Review File has been amended to remove third-party material where no permission to publish could be obtained.

We chose 3 weeks post infection (wpi) as terminal time-point for most of the *in vivo* experiments because we wanted to make sure there is still antigen present in all analysed organs. Chronic LCMV infection without CD4 depletion is being cleared in some tissues already after a month; previous studies (Fuller et al., 2004, Wherry et al., 2003) published viral kinetics without CD4 depletion.

Editorial Note: Figure 3 in this Peer Review File has been amended to remove third-party material where no permission to publish could be obtained.

We have confirmed that mice had a persistent infection by assessing viral titers at 2, 3, and 4 weeks into chronic LCMV infection (Fig. 4a, below). Additionally, since most experiments were performed 21 dpi, we made sure the mice had replicating virus in all analyzed organs (Fig. 4b, below).

Fig. 4: Viral titers. CD45.2⁺ wild-type mice received 10000 CD45.1⁺ P14 cells one day prior infection with a high dose of LCMV clone 13 (2×10^6 ffu/mouse). **a** Viral titers in kidneys isolated from chronically infected mice 2, 3 or 4 weeks into chronic LCMV. **b** 21 days post-infection, mice were sacrificed, blood was drawn from heart, then perfused with 20 mL ice-cold PBS, and organs were harvested: 4 lymph nodes (LN) (inguinal and axillary), the whole lung, 1 lobe of the liver, 1 kidney, the spleen, and the bone marrow (BM) from 1 leg (femur and tibia) was flushed with ice-cold PBS. The harvested organs were weighted and viral titers were quantified according to previously described protocols (Battegay et al., 1991).

Furthermore, we have also verified that LCMV antigen was still present 29 dpi (Fig. 5, below).

Fig. 5: In vivo priming of naïve P14 Nr4a1-GFP cells 4 wpi. Naïve P14 *Nr4a1*-GFP cells were transferred into naïve or chronically infected (29 dpi) hosts. 12 hours after transfer, spleens were harvested. FACS plots show representative examples ($n=2-3$) of *Nr4a1*-GFP and PD-1 expression in P14 cells (gated on alive singlets CD8⁺CD45.1⁺ cells).

These data confirm that mice were indeed chronically infected.

2) Given that TCR signaling is often associated with a calcium flux, it was surprising that such measurements were not described.

A previous study (Agnellini et al., 2007) from our lab has shown that Ca²⁺ influx, triggered by TCR MHC/peptide interactions on APCs, is largely functional in CD8⁺ T cells from chronically infected

mice when exposed to “naïve” target cells (i.e. target cells that have not been isolated from chronically infected hosts)

Editorial Note: Figure 6 in this Peer Review File has been amended to remove third-party material where no permission to publish could be obtained.

3) The authors describe VL4 staining as a readout for virally infected cells, but do not confirm that the cells indeed contain infectious viral particles. While this is very likely correct, confirmation of this assay by direct measurement of virus should be performed.

We have provided viral titer measurements in Fig.1 (in this document). The focus forming assay protocol (Battegay et al., 1991) used to determine LCMV viral titers is based on VL4 staining with the same antibody we used in the FACS analysis (Fig. 5 in the revised manuscript). The advantage of performing a focus forming assay (over *ex vivo* staining) is detecting whether the samples have virus particles capable of active replication. Since the VL4 staining is performed intracellularly, we are not able to isolate VL4⁺ cells to perform qPCR or a focus forming assay. However, it is more likely that VL4^{hi} cells contain active replicating virus than VL4^{neg} because i) the nucleoprotein (stained by the VL4 antibody) is required for functional viral particles and ii) the same antibody clone was used to assess viral titers (assessed by focus forming assay) and for FACS staining.

Reviewer #2 (Remarks to the Author):

CD45.1+ P14 CD8 T cells specific for the LCMV-derived gp33-41 peptide were adoptively transferred into CD45.2+ hosts one day prior infection with a low (200 ffu) or high (2x10⁶ ffu) dose of LCMV clone 13 to induce acute and chronic infections, respectively. Two weeks later, P14 cells were isolated from the spleens and restimulated in vitro with plate-bound α-CD3 and α-CD28 for 4 h and their transcriptional profile defined using bulk RNAseq. Cells isolated from chronically infected mice have a distinct transcriptional profile compared to the ones isolated from acutely infected mice. Upon in vitro stimulation, 429 genes showed similar changes in expression in both CD8 T cells isolated from acutely and chronically infected mice. Only 35 genes were differentially expressed in exhausted cells when compared to effector/memory cells after stimulation. CD45.1+ P14-Nr4a1-GFP CD8 T cells were adoptively transferred into wild type CD45.2+ C57BL/6J mice one day prior infection with a high dose of 2x10⁶ ffu LCMV clone 13. One day after chronic infection the Nr4a1-GFP reporter and PD-1 inhibitory receptor were upregulated and then rapidly downregulated in cells isolated from spleen and various infected tissues. P14-Nr4a1-GFP cells isolated three weeks after chronic or acute LCMV infection and stimulated in vitro with anti-CD3 and anti-CD28 antibodies for 6 hours upregulated the

Nr4a1-GFP reporter. However, Nr4a1-GFP signals from cells isolated from chronically infected mice were lower than those of T cells isolated from acutely infected mice. Antigen was present in CD45.2+ C57BL/6J recipients three weeks after chronic LCMV infection and cannot thus account for the lack of TCR signaling in vivo (at least using the Nr4a1-GFP reporter). Considering that PD-1 is expressed on exhausted P14-Nr4a1-GFP T cell and that PDI-L1 is expressed on infected cells three weeks after chronic LCMV infection, α -PD-L1 was administered to those mice and P14-Nr4a1-GFP cells isolated from various tissues were analyzed three hours after administration. approximately a quarter of P14-Nr4a1-GFP cells expressed some weak levels Nr4a1-GFP signals. Addition of antibodies against other inhibitory receptors (LAG-3, TIM-3, CTLA4, 2B4, TIGIT) did not increase the Nr4a1-GFP signals or the frequency of responding cells. Finally, gp33-41-pulsed target cells originating from naïve mice were transferred into recipients with established chronic LCMV infection and it was analyzed whether their killing would be associated with in vivo TCR signaling in exhausted P14-Nr4a1-GFP T cells. As compared to control mice that received unpulsed target cells, 10% P14-Nr4a1-GFP cells expressed Nr4a1-GFP signals. The elegant and novel data presented in this paper are fairly discussed and in particular the observation that among the screened blocking antibodies PD-1 is the most efficient to rescue Nr4a1-GFP signals. However, most of the data are shown using histograms with % and MFI and as emphasized in Specific Comments it prevents proper evaluation of the data.

We have changed the graphs accordingly in the revised manuscript and provided excel table with raw data.

Specific issues.

1/ The paper is sometimes difficult to read and some words are likely missing in the following sentences: 'Previous studies showed that virus-specific CD8 T cells are transcriptionally distinct when compared to virus-specific CD8 T cells isolated upon acute infection and fail to exert several effector functions such as production of the inflammatory cytokines IFN- γ and TNF- α .' And 'TCR transgenic CD45.1+ P14 CD8 T cells that recognize the LCMV-derived gp33-41 peptide were adoptively transferred into CD45.2+ hosts one day prior infection with a low (200 ffu) or high (2x10⁶ ffu) dose of LCMV clone 13 to induce an acute, respectively chronic infection.'

We have accordingly rephrased the text. (rows 47-49 and 52-54 in the revised manuscript).

2/ Work by the group of Hogquist (PMID: 25137456) shows that T cells express basal /homeostatic levels of Nr4a1-EGFP reporter. Accordingly, a negative control should be shown in Fig. S1 corresponding to P14 CD8 T cells lacking the Nr4a1-EGFP reporter.

We added a negative control in Supplementary Fig. 1.

3/ Fig. 2. A *Nr4a1*-GFP vs PD-1 dot plot corresponding to CD45.1+ P14-*Nr4a1*-GFP CD8 T cells from non-infected mice need to be shown.

We have added a FACS plot from an uninfected mouse in Fig. 2a.

4/ Fig. 3. Is the % of CD107-positive cells similar in the two experimental conditions?

We consistently see largely preserved degranulation in exhausted cells, whether we use antibodies, peptide, or pulsed DCs. These results corroborate previously published data (Agnellini et al., 2007).

Editorial Note: Figure 7 in this Peer Review File has been amended to remove third-party material where no permission to publish could be obtained.

Editorial Note: Figure 8 in this Peer Review File has been amended to remove third-party material where no permission to publish could be obtained.

5/ Fig. S3C. FACS profiles and not only histograms displaying % expression need to be shown for PD-1, CD39, and TIM-3 at 2 to 4 weeks post-infection.

We added FACS plots showing PD-1 (Supplementary Fig. 1), CD39 and TIM-3 expression (Supplementary Fig. 3).

6/ Fig. 7D. Plots similar to the one shown in 6B need to be shown.

We changed the layout of the graphs accordingly in all figures (main text and supplementary).

7/ Fig. 7F. FACS profiles need to be shown.

We added FACS plots.

8/ Are P14-*Nr4a1*-GFP CD8 T cells found in established chronic LCMV infection expressing *BTLA*?

The RNAseq data (Figure 1) suggests that *Btla* is induced by stimulation (in both acute and chronic conditions). There is no significant difference between acute and chronic conditions in unstimulated samples (adjusted p value = 0.745).

Regarding BTLA expression at the protein level, it has been shown that exhausted virus-specific CD8 T cells express significantly lower levels of BTLA than naïve or memory cells (Blackburn et al., 2009) (Fig. 9, below).

Editorial Note: Figure 9 in this Peer Review File has been amended to remove third-party material where no permission to publish could be obtained.

References

- AGNELLINI, P., WOLINT, P., REHR, M., CAHENZLI, J., KARRER, U. & OXENIUS, A. 2007. Impaired NFAT nuclear translocation results in split exhaustion of virus-specific CD8+ T cell functions during chronic viral infection. *Proc Natl Acad Sci U S A*, 104, 4565-70.
- BATTEGAY, M., COOPER, S., ALTHAGE, A., BANZIGER, J., HENGARTNER, H. & ZINKERNAGEL, R. M. 1991. Quantification of lymphocytic choriomeningitis virus with an immunological focus assay in 24- or 96-well plates. *J Virol Methods*, 33, 191-8.
- BLACKBURN, S. D., SHIN, H., HAINING, W. N., ZOU, T., WORKMAN, C. J., POLLEY, A., BETTS, M. R., FREEMAN, G. J., VIGNALI, D. A. & WHERRY, E. J. 2009. Coregulation of CD8+ T cell exhaustion by multiple inhibitory receptors during chronic viral infection. *Nat Immunol*, 10, 29-37.
- FULLER, M. J., KHANOLKAR, A., TEBO, A. E. & ZAJAC, A. J. 2004. Maintenance, loss, and resurgence of T cell responses during acute, protracted, and chronic viral infections. *J Immunol*, 172, 4204-14.
- WHERRY, E. J., BLATTMAN, J. N., MURALI-KRISHNA, K., VAN DER MOST, R. & AHMED, R. 2003. Viral Persistence Alters CD8 T-Cell Immunodominance and Tissue Distribution and Results in Distinct Stages of Functional Impairment. *Journal of Virology*, 77, 4911-4927.
- WHERRY, E. J., HA, S. J., KAECH, S. M., HAINING, W. N., SARKAR, S., KALIA, V., SUBRAMANIAM, S., BLATTMAN, J. N., BARBER, D. L. & AHMED, R. 2007. Molecular signature of CD8+ T cell exhaustion during chronic viral infection. *Immunity*, 27, 670-84.

REVIEWERS' COMMENTS:

Reviewer #1 (Remarks to the Author):

The authors have adequately addressed the previously raised concerns through the addition of new data or editorial modifications. It was not clear where some of the new figures provided to the reviewers were going to be added to the current manuscript. Specifically the Cx3cr1 subsetting cytotoxicity assay was not apparent in the revised manuscript.

Reviewer #2 (Remarks to the Author):

None

REVIEWERS' COMMENTS:

Reviewer #1 (Remarks to the Author):

The authors have adequately addressed the previously raised concerns through the addition of new data or editorial modifications. It was not clear where some of the new figures provided to the reviewers were going to be added to the current manuscript. Specifically the Cx3cr1 subsetting cytotoxicity assay was not apparent in the revised manuscript.

We have now included the cytotoxicity assay (Supplementary Figure 7 in the revised manuscript).

Reviewer #2 (Remarks to the Author):

None